

# A mechanism for biogenic production and emission of MEK from MVK decoupled from isoprene biosynthesis

Luca Cappellin[1,2], Francesco Loreto[3], Franco Biasioli[1], Paolo Pastore[2], Karena McKinney[4]

[1]Research and Innovation Centre, Fondazione Edmund Mach, S. Michele a/A 38010, Italy
[2]Dipartimento di Scienze Chimiche, Università degli Studi di Padova, Padova, Italy
[3]National Research Council, Department of Biology, Agriculture and Food Science (DISBA), Rome 7-00185, Italy
[4]Colby College, Waterville, Maine, USA

*Correspondence to*: L. Cappellin (luca.cappellin@gmail.com)

**Abstract.**

Methyl ethyl ketone (MEK) is an important compound in atmospheric chemistry. While attention has been paid mostly to anthropogenic sources of MEK, recently it has been shown that biogenic sources are globally as important as anthropogenic ones. However, the origin of biogenic MEK has yet to be completely elucidated. We present the full mechanism by which within-plant transformation of MVK and, to a minor extent, of 2-butanol and 3-buten-2-ol, is a source of biogenic MEK. Such

transformation is observed in red oak for both exogenous MVK, taken up from the atmosphere, and endogenous MVK generated within plant upon stress (e.g. heat stress). Endogenous MVK emitted by plants is typically explained by within-plant oxidation of isoprene caused by oxidative stress. In this study we show that MVK and MEK emission caused by heat stress is not related to isoprene in isoprene-emitting plants, implying that the massive carbon investment that plants commit to isoprene production is not explained by a direct antioxidant role. The presented mechanism can be important for inclusion in plant

emission and in plant-atmosphere interaction models.

**Keywords**

Isoprene; oxidative stress; heat stress; within-plant isoprene oxidation; methyl ethyl ketone, methyl vinyl ketone ketone, 2-butanol; 3-buten-2-ol; $^{13}CO_2$ labelling.

**Introduction**

Isoprene is the major biogenic volatile organic compound (VOC) emitted by the biosphere into the atmosphere. A wide breadth of research, and particularly the latest studies employing genetic engineering to produce transgenic plants with modified isoprene emission, has demonstrated that isoprene increases plant tolerance to oxidative stresses, caused by factors such as high temperature and ozone (Behnke et al., 2007; Loivamäki et al., 2007; Sasaki et al., 2007; Sharkey et al., 2005; Vickers et

al., 2009, 2011). However, the mechanisms by which isoprene confers resistance against oxidative stresses remain unclear. A





first hypothesis is that isoprene strengthens thylakoid membranes (Velikova et al., 2011). It has been shown that under physiological conditions there are less than a hundred isoprene molecules per million lipid molecules in the membranes (Harvey et al., 2015), casting doubt on this mechanism. However, there is plenty of evidence that isoprene positively influence the photosynthetic electron flow, reducing heat dissipation mechanisms and stabilizing thylakoid membranes, especially under

stressful environments (Pollastri et al., 2014; Velikova et al., 2011). Recently, it has been shown that isoprene may act as a signaling molecule, inducing up-regulation of phenylpropanoid biosynthetic genes, indirectly enhancing plant resistance to heat and light stress (Harvey and Sharkey, 2016). Alternatively, isoprene may simply act as a proxy of induced activation of secondary metabolism (Tattini et al., 2015). A last hypothesis is that isoprene scavenges reactive oxygen species (ROS) by direct or indirect reaction (Loreto and Velikova, 2001). ROS exert an important signaling role in plants (Mittler et al., 2011),

but can be generated in high amounts upon abiotic stresses, leading to cell damage and programed cell death (Delledonne et al., 2001). Isoprene biosynthesis has been ascribed a ROS quenching effect, but, again, isoprene concentration in the leaf may be too low; abundant lipids and carotenoids can react with ROS much more rapidly than isoprene (Harvey et al., 2015). Nevertheless, emission of putative isoprene oxidation products that may be formed upon isoprene reaction with ROS has been reported, e.g. in a tropical rainforest mesocosm (Jardine et al., 2012), in mango trees (Jardine et al., 2012, 2013) and creosote

bush (Jardine et al., 2010). The putative isoprene oxidation products, the carbonyls methyl vinyl ketone (MVK) and methacrolein (MACR), are cytotoxic and must be rapidly removed from leaves once formed (Oikawa and Lerdau, 2013; Vollenweider et al., 2000). Plants also rapidly take up MACR and MVK (Andreae et al., 2002; Karl et al., 2004, 2005, 2010), suggesting that these compounds may be metabolized. Detoxification mechanisms for MACR have been reported (Muramoto et al., 2015), whereas this is not the case for MVK. The existence of in-plant detoxification mechanisms for these oxidation

products supports the hypothesis that isoprene acts directly as an antioxidant.

Methyl ethyl ketone (MEK) is an important oxygenated volatile organic compound (OVOC) for the atmosphere. In several studies the MEK concentration in the free troposphere has been reported to be approximately 25% of the concentration of acetone (Moore et al., 2012; Singh, 2004). However, MEK has a much shorter lifetime than acetone due to a much higher reactivity (by about one order of magnitude) with hydroxyl radical (OH, $k_{OH}$ = 1.15·$10^{-12}$ cm³ s⁻¹ at 296 K (Chew and Atkinson,

1996)), which makes it important to total OH reactivity (Nölscher et al., 2016). Both biogenic and anthropogenic sources contribute to global atmospheric MEK (Yáñez-Serrano et al., 2016). In urban environments, MEK is typically the most abundant ketone after acetone (Feng et al., 2005; Grosjean et al., 1996), as it is widely produced in industrial processes (Legreid et al., 2007; Sin et al., 2001). The regional background of MEK was once attributed to anthropogenic sources only. More recent findings suggest that the biogenic source of MEK is comparable to the anthropogenic source (Yáñez-Serrano et al., 2015,

2016), but the origin of biogenic MEK has not been fully elucidated so far. Direct emission of MEK from vegetation seems to be the largest contribution since only a few minor biogenic VOCs (e.g, n-butane and 2-butanol) can lead to MEK via atmospheric oxidation (de Gouw et al., 2003; Jenkin et al., 1997; Singh, 2004; Sommariva et al., 2011). We recently found a possible relation between the detoxification by vegetation of MVK and the biogenic emission of MEK (Cappellin et al., 2017).



We suggested that MVK could be efficiently detoxified by reduction reactions which lead mostly to MEK and, to a minor extent to 2-butanol, though the full mechanism was not described.

In this study we explore the links between production of isoprene from photosynthetic carbon, within-plant isoprene oxidation, and the biogenic emission of putative isoprene oxidation products MVK, MEK, and 2-butanol. We performed a series of fumigation experiments in which leaves were exposed to exogenous levels of a single carbonyl, and investigated the relationships between the uptake of each compound and the release of related products. We then studied emission of carbonyls by leaves under progressively higher temperature conditions that likely inhibit photosynthesis and increase isoprene biosynthesis and the oxidative environment within leaves. We finally explored whether MVK, MEK, and 2-butenol produced under heat stress are products of in-plant isoprene oxidation by determining whether they share a labeling pattern with isoprene when leaves are fed $^{13}CO_2$. Different labeling patterns would indicate that emitted carbonyls are not isoprene oxidation products, and would cast doubt on the hypothesis that isoprene acts as an antioxidant by reacting with ROS inside leaves.

## Results

**Fumigation of leaves with OVOCs.** Individual plants were fumigated with MVK, MEK, 2-butanol, 3-buten-e-ol, or MACR, as described in the Materials and Methods (Figure 1). Both MVK and MACR were efficiently taken up by the fumigated leaves but, while MVK was continuously transformed into other volatile products that were subsequently released (Figure 1a), in the case of MACR only a transient release of other volatiles was observed (Figure 1e). MVK reduction generated either MEK or 3-buten-2-ol, depending on whether the alkene moiety or the carbonyl moiety of MVK is reduced. Further reduction reactions converted MEK and 3-buten-2-ol into 2-butanol (see also Figures 1b and 1d). The release of MEK, 3-buten-2-ol, and 2-butanol corresponded to $97 \pm 6\%$ (mean ± std, n=3) of the total MVK uptake. In the MEK fumigation experiments (Figure 1b), the release of 2-butanol corresponded to $87 \pm 22\%$ of the total uptake of MEK. Remarkably, 2-butenol and 3-buten-2-ol may also convert into MEK (Fig. 1c, d). In the 2-butanol fumigation experiments (Figure 1c), the release of MEK corresponded to $90 \pm 50\%$ of the total uptake of 2-butanol. In the 3-buten-2-ol fumigation experiments (Figure 1d), the release of MEK and 2-butanol corresponded to $56 \pm 18\%$ and $22 \pm 6\%$ of the total uptake of 3-buten-2-ol, respectively. Finally, in the MACR fumigation experiment (Figure 1e) we observed a small transient release of isobutyraldehyde, in an amount accounting for $6.5 \pm 1.3\%$ of the MACR taken up, and, to a lesser extent, of 2-methallyl alcohol ($0.7 \pm 0.5\%$) and isobutanol ($2.1 \pm 0.5\%$). The total emission of these three compounds corresponded to $9.3 \pm 1.4\%$ of the total MACR taken up.

**Emission of isoprene and OVOCs under heat stress.** Emissions from darkened, unstressed leaves were measured (Fig. 2a). As expected (Loreto and Sharkey, 1990), a very small basal emission of isoprene ($0.02\pm0.01$ nmol m$^{-2}$ s$^{-1}$) was found. In these leaves the emissions of putative isoprene oxidation products (MVK+MACR+ISOPOOH, Fig. 2b) and their reduction products (MEK and 2-butanol. Fig. 2c and d, respectively) were near background level, and dark respiration was low ($0.5\pm0.2$ μmol m$^{-}$



$^2$ s$^{-1}$, Fig. 2e). When darkened red oak leaves were exposed to mild heat-stress (35°C for 2 h) a small but statistically significant increase in isoprene emission to 0.05±0.01 nmol m$^{-2}$ s$^{-1}$ was observed. Emission of MVK+MACR+ISOPOOH was detected at a level of 0.001±0.002 nmol m$^{-2}$ s$^{-1}$, and emissions of further transformation products of MVK were also detected, namely MEK (0.006±0.004 nmol m$^{-2}$ s$^{-1}$) and 2-butanol (0.010±0.001 nmol m$^{-2}$ s$^{-1}$). However, only the increase of 2-butanol was

statistically significant compared to unstressed leaves. Dark respiration also increased to 0.8±0.4 μmol m$^{-2}$ s$^{-1}$. More severe heat stress was then imposed, maintaining the darkened leaves at 45°C for 2 h. This led to a further significant increase in isoprene emission (0.10±0.03 nmol m$^{-2}$ s$^{-1}$) (Fig. 2a). Emission of MVK+MACR+ISOPOOH also increased to 0.006±0.003 nmol m$^{-2}$ s$^{-1}$ (Fig. 2b) and emission of MEK and 2-butanol, increased to 0.03±0.01 nmol m$^{-2}$ s$^{-1}$ and 0.04±0.02 nmol m$^{-2}$ s$^{-1}$, respectively (Fig. 2c, d). Dark respiration also increased to 1.1±0.2 μmol m$^{-2}$ s$^{-1}$ (Fig. 2e). All these changes were statistically

significant with respect to the values measured at 25°C. After a recovery phase of 2 h at room temperature (25°C), pre-stress emissions were again found for all compounds (Fig. 2a-e).

The same experiment was repeated on illuminated red oak leaves (Fig. 2f-j). Non-stressed leaves (maintained at 25 °C) photosynthesized at a rate of 1.1±0.5 μmol CO$_2$ m$^{-2}$ s$^{-1}$ (Fig. 2j) and emitted isoprene at a rate of 4±2 nmol m$^{-2}$ s$^{-1}$ (Fig. 3f). These values for carbon assimilation and isoprene emission are consistent with the light intensity to which plants were exposed

(e.g. see (Loreto and Sharkey, 1990)). In non-stressed leaves, the emission of MVK+MACR+ISOPOOH was below the detection limit (Fig. 2g) and the emissions of MEK and 2-butanol were also close to background (Fig. 2h, i). Exposure to 35°C for 2 h led to a non-significant decrease of photosynthesis (Fig. 2j) and to a large increase of isoprene emission, reaching 10±5 nmol m$^{-2}$ s$^{-1}$ (Fig. 2f). Detectable emission rates of MVK+MACR+ISOPOOH were present (Fig. 2g), and MEK and 2-butanol emissions were also detectable (Fig. 2h, i). However, the change in emissions compared to that observed at 25°C was

statistically significant only for 2-butanol. Severe stress (45°C for 2 h) caused a reduction in the emission of isoprene as compared to the maximum emission observed at 35°C (Fig. 2f), and photosynthesis was also inhibited (Fig. 2j). However, the emission of MVK+MACR+ISOPOOH increased to 0.06±0.06 nmol m$^{-2}$ s$^{-1}$ (Fig. 2g), and MEK and 2-butanol increased to 0.03±0.01 nmol m$^{-2}$ s$^{-1}$ and 0.03±0.01 nmol m$^{-2}$ s$^{-1}$, respectively (Fig. 2h, i). After recovering for 2 h at room temperature (25°C), VOC emissions and photosynthetic rate were similar to those of pre-stressed leaves (Fig. 2f-j).


**$^{13}$C labelling of isoprene and OVOCs.** To provide better insights about isoprene production and possible within-leaf oxidation under severe heat stress, plants were exposed to isotopically labeled CO$_2$ at 45°C. $^{13}$C labelling of isoprene was clearly visible. All $^{13}$C-labeled isoprene isotopomers (m/z 70-74) appeared, and the unlabeled isoprene (m/z 69) declined rapidly (Fig. 3). After 20 min of $^{13}$CO$_2$ labelling, the percentage of labelled isoprene was above 80%, and a stable labeled fraction of 95±1 %

of the emitted isoprene was reached ca. 40 min after starting the $^{13}$CO$_2$ fumigation (Fig. 4a). However, MVK+MACR+ISOPOOH, MEK and 2-butanol were not labelled by $^{13}$C. Even after 2 h of $^{13}$CO$_2$ labelling, the isotopic ratios of MVK+MACR+ISOPOOH (Fig. 4b) and those of MEK and 2-butanol (Fig. 4c, d) remained at their natural abundance levels.



## Discussion

There are consistent indications that MEK is released by vegetation at the leaf-, canopy-, and ecosystem-scale and that biogenic sources influence global atmospheric concentrations of MEK ((Yáñez-Serrano et al., 2016) and references therein). However,

no mechanism for biogenic MEK production has been demonstrated previously. It has been hypothesized that MEK could be generated following similar pathways to acetone, as a by-product of cyanogenesis in the transformation of a cyanohydrin lyase (Yáñez-Serrano et al., 2016). This idea has not been proven and emission would be limited to cyanogenic plants. Plant uptake of atmospheric MVK is a well-established phenomenon (Karl et al., 2010) and bidirectional exchange of MVK has been sometimes reported (Karl et al., 2005). Emission of MVK and MEK has been rarely measured at leaf level and is challenging

to measure because i) emission is small, especially in the case of MVK; ii) separation of leaf and atmospheric sources in the presence of high isoprene emissions is difficult. Jardine and co-workers (Jardine et al., 2012, 2013) reported emission of methacrolein (MACR) and methyl vinyl ketone (MVK) in isoprene emitting trees. They attributed carbonyl emissions to within-leaf isoprene oxidation, possibly by ROS induced under environmental stress conditions (e.g. heat stress).

Our results confirm emission of carbonyls and show that reduction of several carbonyls may occur in plants. In particular, we show that reduction reactions transform MVK into MEK, 3-buten-2-ol, and 2-butanol, and that the conversion of MEK into 2-butanol occurs in both directions (Fig. 5). This mechanism is the first to explain biogenic production and emission of MEK. The results also suggest that isoprene oxidation is not the source of these VOCs.

Our fumigation experiments establish a causal link between plant metabolism of MVK and MEK production. Reaction yield calculations were performed considering the net total amount of MVK taken up during the experiment and the net total amount of MEK released. We show that 73 ± 10% of the MVK taken up by leaves was converted into MEK. Most of the remaining MVK was converted into 3-buten-2-ol and 2-butanol. Thus 97 ± 6% of MVK uptake was re-emitted as other volatile products. These data confirm previous reports at Harvard Forest, corroborated by laboratory experiments, where a correlation between

uptake of MVK and emission of MEK, 2-butanol and 3-buten2-ol was found (Cappellin et al., 2017). The implication is that metabolism of MVK is a likely source of MEK emissions.

Whether other mechanisms for MEK production by plants exist remains an open question. Processes on the leaf surface such as MVK adsorption or metabolization by leaf surface bacteria have been shown to be of minor importance in previous

investigations of ketone and aldehyde leaf uptake (Omasa et al., 2000; Tani et al., 2010, 2013; Tani and Hewitt, 2009, p.20). Future studies should investigate the fraction of global biogenic MEK emissions that could be accounted for by the proposed production mechanism.



Rapid transformation of MVK by secondary reactions can be interpreted as a necessary and efficient detoxification mechanism. MVK toxic effects include accumulation of hydrogen peroxide and the activation of some stress-responsive genes (Alméras et al., 2003; Vollenweider et al., 2000). MVK metabolism has been reported in several plants (Karl et al., 2010; Tani et al., 2010). It has also been demonstrated that various plant cells are able to reduce unsaturated ketones (Kergomard et al., 1988).

Long-term fumigation experiments with houseplants (Tani and Hewitt, 2009) have established that the uptake of several aldehydes and ketones exceeded by orders of magnitude the amount dissolved in leaf water, thus implying their metabolization as well. Transformation of MVK into MEK may be especially important in stressed plants as oxidative stress increases and endogenous production of MVK occurs (Jardine et al., 2012). The fumigation experiments suggest that the process can occur for exogenous MVK as well. Hence, the reduction mechanism also provides an explanation for plant uptake of MVK, a

cytotoxic compound, which has been observed in many studies (e.g. (Brilli et al., 2016; Karl et al., 2005, 2010)).

MACR was also efficiently metabolized, consistent with other literature reports on tomato plants (Muramoto et al., 2015). In this case, however, detected volatiles produced by MACR reduction represented only about 9% of the MACR uptake. By analogy with the case of MVK, the expected reduction products of MACR would be isobutyraldehyde, 2-methylprop-2-en-

1-ol, and isobutyl alcohol. However, only a transient release of such products was detected. Isobutyraldehyde emission accounted for 6.5% of the MACR uptake within two hours, confirming results from a previous study on tomato plants (6.4% (Muramoto et al., 2015)). Muramoto *et al.* (Muramoto et al., 2015) reported that most MACR in the leaf undergoes glutathionylation. Therefore, no MACR volatile products are released. Our results are in line with this interpretation, as MACR uptake implies a sink within leaves, suggesting a continuous metabolization of the compound, but the metabolic products are

not released as volatiles.

The foliar uptake of MVK and MACR was rapid and sustained throughout the entire fumigation period, suggesting a large and fast metabolic sink. On the other hand, the uptake of secondary transformation products (namely MEK, 3-buten-2-ol, and 2-butanol) was not sustained, indicating that the metabolic sink was smaller and insufficient to rapidly scavenge the metabolites,

leading to their temporary accumulation within leaves. Upon ending the fumigation, accumulated MEK, 3-buten-2-ol, and 2-butanol were released from the leaves back to the atmosphere (Fig. 2b, 2c, 2d).

A further aim of this study was to verify whether carbonyls emitted by plants come from isoprene oxidation in plants, as elsewhere proposed (Jardine et al., 2012). The experiments were conducted on red oak, one of the strongest isoprene-emitting

plant species (Loreto and Sharkey, 1990). As isoprene biosynthesis is light- and temperature-dependent (Loreto and Sharkey, 1990), an increase of isoprene transformation products would be expected when leaves are illuminated and exposed to moderately high temperatures. This behavior was indeed represented in the present data (Fig. 2). Under heat stress, photosynthesis and isoprene emission generally decouple and carbon sources other than direct products of photosynthetic carbon fixation are used for isoprene biosynthesis (Brilli et al., 2007). When leaves were severely heat-stressed we did not





observe the expected uncoupling between photosynthesis and isoprene emission, as both parameters decreased. However, MVK+MACR+ISOPOOH, MEK and 2-butanol increased. This increase would be predicted to occur under an increasingly oxidative environment where molecular $O_2$ is directly photoreduced instead of being used for photosynthetic electron transport. Interestingly, the amount of MVK+MACR+ISOPOOH, MEK and 2-butanol was similarly enhanced in illuminated and

darkened leaves under the effect of high temperatures, despite the fact that no photosynthesis and minimal isoprene emission was present in darkness.

This unexpected observation was followed up with measurements of $^{13}C$ labeling to further characterize the carbon source of MVK, MEK, and 2-butanol. As photosynthetic metabolites and isoprene label very quickly and almost totally when exposed

to a $^{13}CO_2$ atmosphere ( (Delwiche and Sharkey, 1993), see also Figs 3, 4), we would expect that putative isoprene oxidation products and their further transformation products would be quickly and totally labeled as well. On the other hand, if carbon sources other than direct photosynthetic metabolites (including isoprene) are used for their biosynthesis these compounds would not be labeled. The observation that no $^{13}C$ labeling appeared in any of the emitted carbonyls suggests that isoprene is not involved in their biosynthesis. We are not aware of isoprene pools that label slowly in plants (Niinemets et al., 2004).

However, sources independent of freshly assimilated carbon contribute to isoprene production when photosynthesis is stress-constrained (Brilli et al., 2007). Photorespiratory carbon may contribute to isoprene production especially during dynamic changes of environmental conditions (Jardine et al., 2014). However, photorespiration also labels quite rapidly with $^{13}CO_2$ (Delfine et al., 1999), and $^{13}C$ would appear in MVK and MEK skeletons, if contributed by photorespiratory carbon.

Isoprene labelling experiments using $^{13}C$-enriched glucose under $CO_2$-free air conditions have provided a clear indication that isoprene may also  incorporate carbon via the glycolytic pathway (Affek and Yakir, 2003), and possibly also by re-fixation of respiratory $CO_2$ (Loreto, 2004; Loreto et al., 2007). This alternative carbon source can become prevalent when photosynthesis is stress-inhibited (Brilli et al., 2007), and may sustain small yet measurable emissions of isoprene in $CO_2$-free air (Affek and Yakir, 2002) or in detached leaves having no photosynthetic activity (Brilli et al., 2011; Harrison et al., 2013; Loreto and

Schnitzler, 2010). However, under the conditions used here photosynthesis was never dramatically impaired and isoprene emission was labelled completely, even under heat stress. We conclude that the emitted carbonyls are not isoprene oxidation products but derive from a different metabolism. It should be noted, however, that in nature, under high UV radiation, the emission of isoprene is generally far larger than that found in our experiment, and the oxidation potential of the atmosphere is also greater. Thus, we cannot completely rule out that isoprene also

oxidizes forming carbonyls, namely MVK and MACR, in nature. If not isoprene, then what is the source of the emitted carbonyls in plants? MVK and MACR are peroxidation products of the trienoic fatty acid contained in chloroplast membranes (Alméras et al., 2003; Kai et al., 2012), and cause cytotoxicity once formed in leaves. MVK



also is a component of flower scent (Knudsen et al., 1993) and may be a lipid peroxidation product, activating the expression of stress-related genes in Arabidopsis (Vollenweider et al., 2000). As we have demonstrated that isoprene is not responsible for the production in plants of MVK and its transformation products, further research should concentrate on investigating the relationships between emitted carbonyls and lipid catabolism in cellular
membranes.

## 2 Materials and Methods

### 2.1 Plant material and experimental design

Experiments were performed on two-year-old red oaks (*Quercus rubra*) obtained from a local nursery (Malleier, Lana, Italy). Forty plants were placed in 5 L polypropylene pots (Plastecnic, Perego, Italy), using TerraBrill® peat moss (Agrochimica,
Bolzano, Italy). Prior to the experiments, plants were grown for 60 d in a greenhouse under a 14 h photoperiod with a light:dark temperature regime of 24.0:19.0 °C, $60 \pm 10$ % relative humidity, and ca. 90 µmol $m^{-2} s^{-1}$ light intensity. Plants were watered every three days and developed fully expanded leaves after five weeks. Four days prior to the experiments, plants were transferred from the growth chamber into a climate cabinet (Climacell 707, BMT Medical Technology, Brno, Czech Republic). This early transfer allowed plant recovery from accidental mechanical injures and adaptation to the new environment. The
climate cabinet was set with the same parameters of the greenhouse, but at a constant temperature of 25°C, and interfaced with the PTR-ToF-MS via polyetheretherketone (PEEK) capillary tubes (ca. 1.5 m length x 1.01 mm ID, temperature: 110°C, flow: 40 sccm).

One day prior to the start of the experiments, a shoot portion of each plant was enclosed within a Teflon (perfluoroalkoxy; PFA) bag and three capillary tubes were attached to each shoot to be monitored. The three tubes included the following: a PFA
tube providing a constant flow of humidified zero air (eventually including fumigation VOCs during fumigation experiments) to the VOC-bag, a second PFA tube removing the overflow air, and a PEEK capillary tube sampling the VOC-bag air into the PTR-ToF-MS. As negative control, volatiles were monitored in parallel on an empty VOC-bag connected to the PTR-ToF-MS with the same tubing system described above.

Fumigation experiments were performed on three independent plants and on the empty VOC-bag as a negative control. New
plants were used for every fumigation. Pure liquid standards of each chemical were purchased from Sigma-Aldridge and diluted with Milli-Q water in 1:25,000 ratio. A Liquid Calibration Unit (Ionicon Analytik GmbH, Innsbruck, Austria) was employed to continuously nebulize 10 µL $min^{-1}$ of the diluted standard into 1 L $min^{-1}$ of synthetic air generated from a cylinder (Aria Medicinale F.U., Rivoira) made by 80% $N_2$, 20% $O_2$, and 400 ppm $CO_2$. Fluxes were controlled using mass flow controllers (MKS Instruments, Deutschland GmbH). Water vapour was added to the air stream by constantly nebulizing water
via a Liquid Calibration Unit (Ionicon Analytik GmbH, Innsbruck, Austria) to reach a RH of 60%.



Fumigation experiments were performed separately for each chemical using fresh plants from the greenhouse. Gas flow measurements in the PFA tubes providing the fumigation gas mixture were conducted using a flow meter at the beginning and at the end of each experiment (Defender 530, DryCal Technology, MesaLabs) and was ca. 250 sccm for each VOC-bag. Leaf area enclosed in the bag was measured using an open-source software (ImageJ, available at http://imagej.net/Welcome). VOC

uptake or release by the plants was calculated as the difference between VOC concentrations in the plant VOC-bag and in the empty VOC-bag. Results were normalized by the total leaf area enclosed in the bag, and by the incoming air flows.

Heat stress experiments were conducted using a similar setup than in the fumigation experiments. After acclimation for four days, during the day of the experiment, basal emissions at 25°C were measured for two hours before raising the temperature to 35°C (mild stress) and subsequently to 45°C (severe stress). Each temperature was maintained for two hours. Measurements

were continued during the recovery phase at 25°C for two more hours. Transitions between different temperatures required a 15 minute ramp.

Experiments using labelled $^{13}CO_2$ were performed a follows. A 1-L cylinder of $^{13}C$-labelled $CO_2$ was purchased from Sigma-Aldrich (99 atom % $^{13}C$ purity). The heat stress experiment at 45°C described above was repeated switching the $CO_2$ in the air stream from unlabelled $CO_2$ to $^{13}C$-labelled $CO_2$. The labelling was triggered when the signals for isoprene and isoprene

primary and secondary oxidation products became stable after the temperature change, i.e. about one hour after the onset of the stress. The temperature was kept constant at 45°C while feeding $^{13}C$-labelled $CO_2$. Results were compared to analogous experiments employing unlabelled $CO_2$.

## 2.2 Gas analysis

Trace gas analysis performed via a PTR/SRI-ToF-MS 8000 (Ionicon Analytik GmbH, Innsbruck, Austria). PEEK capillary

tubes directly sampled the air mixture from each VOC-bag to the instrument inlet. A valve system allowed switching between the enclosures every two minutes. The PTR/SRI-ToF-MS was equipped with a switchable reagent ion system (Jordan et al., 2009), allowing to select either $H_3O^+$ or $NO^+$ as primary ion. Measurements were generally taken using $H_3O^+$ ion chemistry, while for the fumigation experiments measurements were taken in both modes in order to strengthen compound identification. Calibrations with pure standards (purchased from Sigma-Aldrich) were carried out for all measured compounds using a Liquid

Calibration Unit (Ionicon Analytik GmbH, Innsbruck Austria). For isoprene, the calibration was performed using a standard gas cylinder (Scott Specialty Gases/Air Liquide) containing isoprene (80.0 ± 5% ppm). Isoprene signal was monitored on the ion peaks, corresponding to proton transfer reactions leading to $C_5H_9^+$ for $H_3O^+$ mode and $C_5H_8^+$, corresponding to charge transfer reactions, for $NO^+$ mode (Karl et al., 2012). The peaks corresponding to the isotopologues were used to monitor the isotopic labelling. Such peaks were $^{13}CC_4H_9^+$, $^{13}C_2C_3H_9^+$, $^{13}C_3C_2H_9^+$, $^{13}C_4CH_9^+$, $^{13}C_5H_9^+$ for $H_3O^+$ mode and $^{13}CC_4H_8^+$,

$^{13}C_2C_3H_8^+$, $^{13}C_3C_2H_8^+$, $^{13}C_4CH_8^+$, $^{13}C_5H_8^+$ for $NO^+$ mode. Ketones undergo proton transfer reactions in $H_3O^+$ mode and three body association reactions in $NO^+$ mode. This was the case for MVK and MEK, leading to the ion signals $C_4H_7O^+$ and $C_4H_9O^+$, respectively, in $H_3O^+$ mode and $C_4H_6O·NO^+$ and $C_4H_8O·NO^+$, respectively, in $NO^+$ mode. Aldehydes react at collision rates via proton transfer in $H_3O^+$ mode and mainly hydride ion transfer in $NO^+$. In $NO^+$ mode aldehydes also undergo three body



association reactions at a much reduced rate (Liu et al., 2013). This is the case of MACR and isobutyraldehyde, mostly leading to the ion signals $C_4H_7O^+$ and $C_4H_9O^+$, respectively, in $H_3O^+$ mode and $C_4H_5O^+$ and $C_4H_7O^+$, respectively, in $NO^+$ mode. Alcohols typically undergo proton transfer followed by dehydration in $H_3O^+$ mode and hydride ion transfer in $NO^+$ mode. Details on the spectral peaks used to monitor each compound in red oak and on the instrument sensitivities are reported

elsewhere (Cappellin et al., 2017, p.2). Isotopic labelling was monitored using the signal corresponding to isotopologues analogously to the case of isoprene. Photosynthesis and dark respiration were measured as $CO_2$ exchange. The exchange of $CO_2$ and $^{13}CO_2$ was estimated in $H_3O^+$ mode using the spectral peaks at m/z 44.9971 and 46.0005, respectively. Since $CO_2$ does not react with $H_3O^+$ at collision rate, extensive calibrations with $CO_2$ and $^{13}CO_2$ standards were made at the experimental conditions. The conditions in the instrument reaction cell were the following: 2.19 mbar drift pressure, 60 °C drift tube

temperature, and 542 V drift voltage for the $H_3O^+$ mode, resulting in an $E/N$ ratio of ca. 125 Townsend (Td) ($N$ corresponding to the gas number density and $E$ to the electric field strength; 1 Td=$10^{-17}$ $Vcm^2$); 2.21 mbar drift pressure, 90 °C drift tube temperature, and 296 V drift voltage in $NO^+$ mode, resulting in $E/N$ ratio of ca. 74 Townsend (Td). The purity of the primary ions was high. In $H_3O^+$ mode the fraction of spurious $NO^+$ and $O_2^+$ ions was <0.3% and <2.1%, respectively. The amount of primary ions was $3.1·10^6$ cps and the signal of the first water cluster was about 11%. In $NO^+$ mode the fraction of spurious

$NO_2^+$, $O_2^+$ and $H_3O^+$ were <3.5 %, <0.2 % and <0.1 % respectively, relative to the primary $NO^+$ ion signal. The amount of primary ions was $2.6·10^6$ cps. The ion detector in PTR/SRI-ToF-MS was a time-of-fight (ToF) mass analyser having mass resolution of about 4000 (m/$\Delta$m, FWHM). The sampling time per bin of ToF acquisition was 0.1 ns, amounting to 350000 bins for a m/z spectrum ranging up to m/z = 400. The spectral ion signals used to derive VOC concentrations were calculated following the procedure described by (Cappellin et al., 2011b). The detector dead time was corrected for applying a procedure

based on the Poisson statistics (Cappellin et al., 2011a; Titzmann et al., 2010). Internal mass calibration was applied achieving a mass accuracy better than 0.001 Th (Cappellin et al., 2010). Extraction of ion counts was performed according to (Cappellin et al., 2011b), employing an optimized peak shape from the sample spectra for fitting.

## 2.3 Statistics

Statistical analyses were performed using R routines (R Development Core Team, 2009) developed in-house. Statistical

differences of emissions or uptakes in different conditions were assessed using Kruskal-Wallis test at a significance level of p<0.05.

**Data availability**

The dataset is available from http://doi.org/10.4121/uuid:63039e2e-03d6-455d-81de-e94c9671f21c.


*Acknowledgements.* Luca Cappellin acknowledges funding from H2020-EU.1.3.2 (grant agreement n. 659315).



*Author contributions*. L. Cappellin, F. Loreto and K. McKinney designed this research. Laboratory analyses were performed by L. Cappellin. L. Cappellin, F. Loreto, F. Biasioli, P. Pastore, and K. McKinney wrote the paper.

*Competing interest*. The authors declare that they have no conflict of interest.

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

**Figure captions**





**Figure 1**. Uptake and transformation of MVK (a) and MACR (e) and of MVK transformation products, MEK (b), 2-butanol (c), 3-buten-e-ol (d) by red oak leaves. Negative values denote uptake, while positive values indicate emission. Black arrows indicate the beginning and the end of the fumigation. In panel (b), the pulses of MEK uptake at the beginning of the fumigation and of MEK release after the end of the fumigation correspond to formation and release, respectively, of a MEK pool dissolved into the leaf water. At equilibrium the uptake of MEK and subsequent release of 2-butanol is constant and indicates a constant rate of MEK transformation within leaves. Analogous considerations can be made for 2-butanol and 3-buten-2-ol in panels (c) and (d), respectively.







5 **Figure 2**. Left: Emission of isoprene (a), MVK+MACR+ISOPOOH (b), products from MVK reduction (MEK (c) and 2-butanol (d)) and respiration (e) upon exposure to moderate (35°C) or severe (45°C) heat stress, and recovery to 25°C by red oak plants in the dark. Asterisks indicate significant differences (Kruskal-Wallis, p<0.05) compared to the unstressed (25°C) case. Right: Emission of isoprene (f), MVK+MACR+ISOPOOH (g), products from MVK reduction (MEK (h) and 2-butanol (i)), and photosynthesis (j) upon exposure to moderate (35°C) or severe (45°C) heat stress, and recovery to 25°C in red oak

10 plants in the light (ca. 90 µmol m$^{-2}$ s$^{-1}$ light intensity). Asterisks indicate significant differences (Kruskal-Wallis, p<0.05) compared to the unstressed (25°C) control.

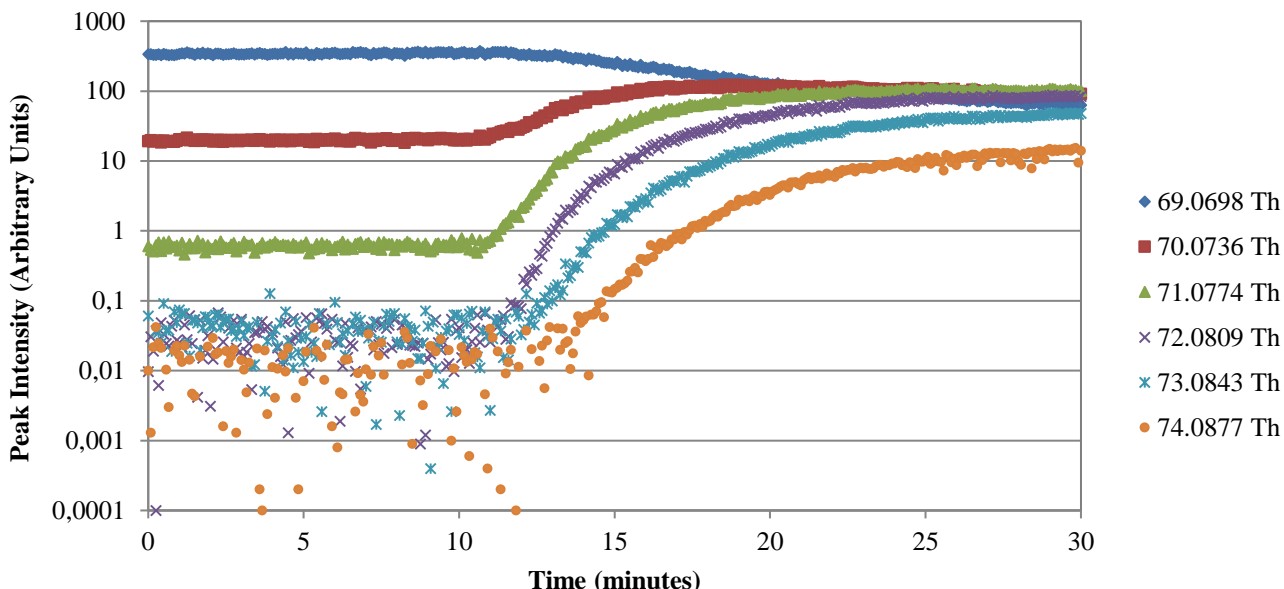

**Figure 3**. Labelling of isoprene emitted by red oak leaves upon $^{13}CO2$-feeding in the air. Fully labelled $^{13}CO_2$ is constantly supplied to the plants starting from time T = 10 min.



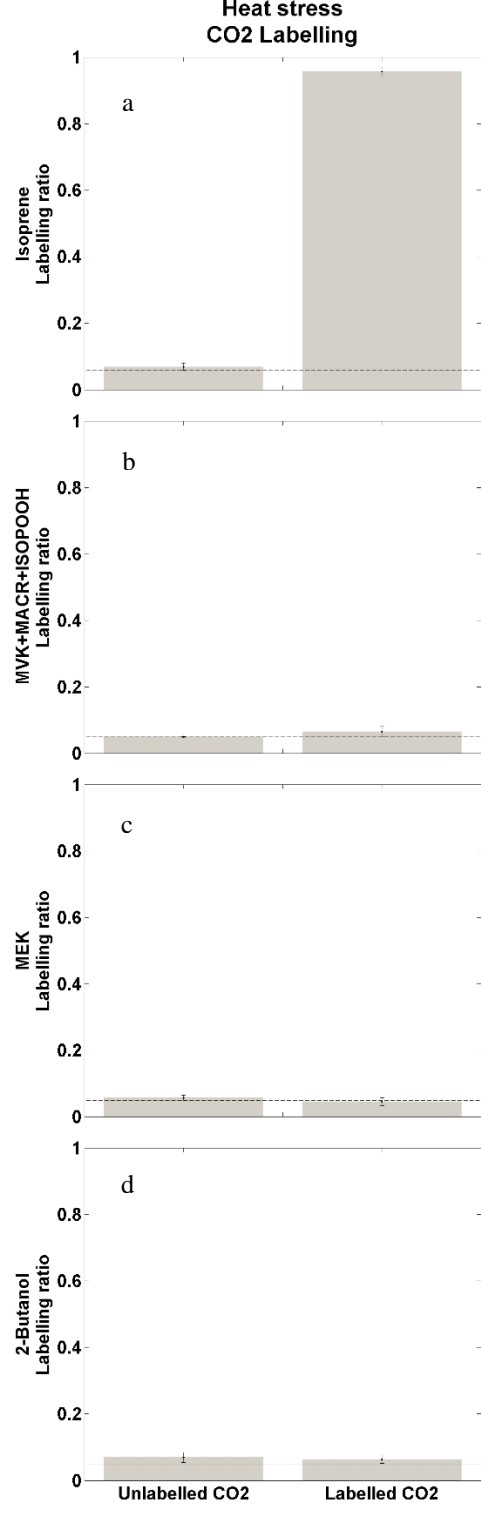



**Figure 4**. Emission of isoprene (a), MVK+MACR+ISOPOOH (b), MEK (c), and 2-butanol (d) in red oak plants heat stressed at 45°C for two hours in the light, under $^{12}CO_2$ or $^{13}CO_2$ atmosphere. Dashed horizontal lines represent natural abundances of isotopic compounds.

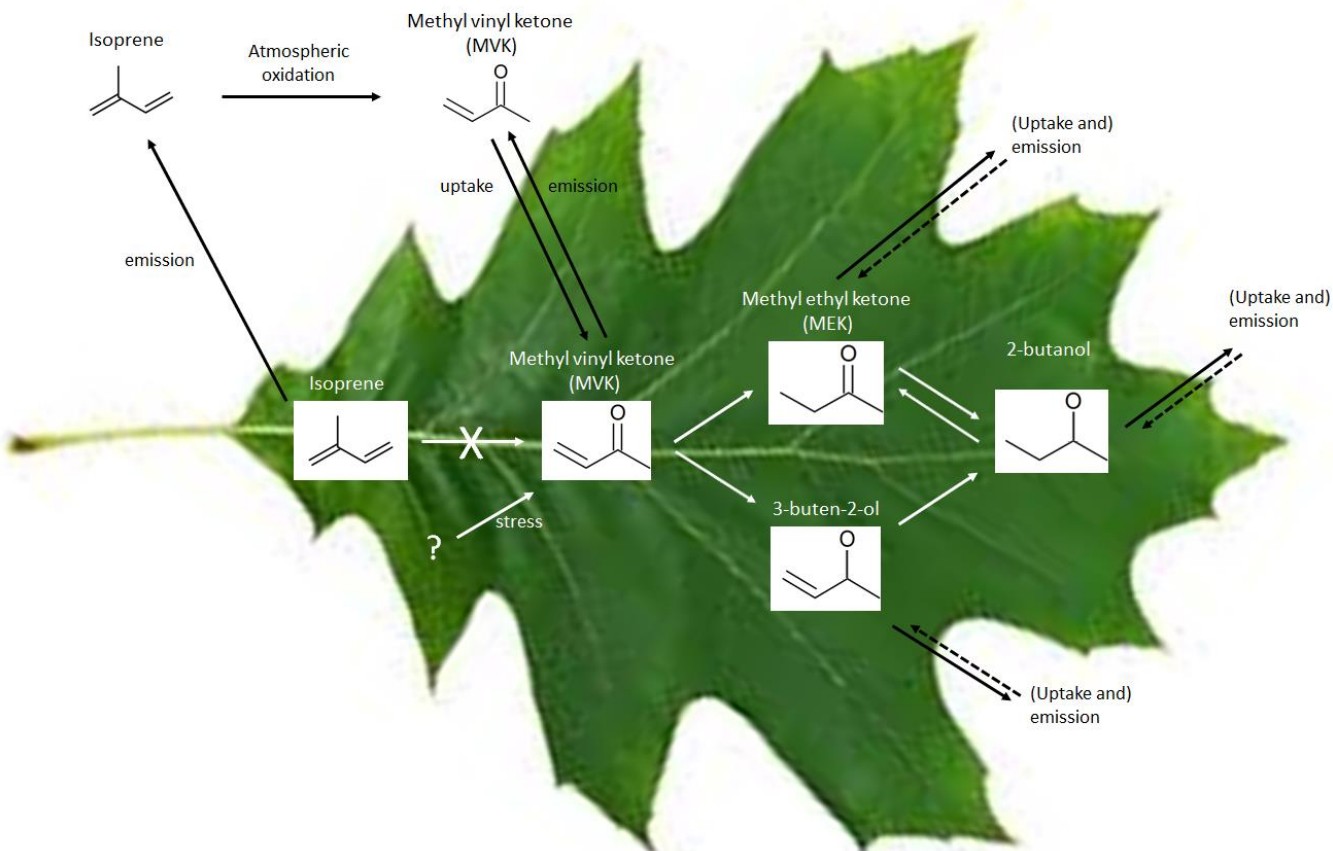

**Figure 5**. Schematics of MVK, MEK, 3-buten-2-ol, and 2 butanol origin and interconversion in leaves, as suggested by this study. The uptake of MEK, 3-buten-2-ol, and 2-butanol is indicated using dotted lines since it has not been reported so far but
15   it is in principles possible given the present results.