# Peer review of "A mechanism for biogenic production and emission of MEK from MVK decoupled from isoprene biosynthesis"

_Atmospheric Chemistry and Physics, 2018_

## Author Comment (AC1) · 2 Dec 2018

Please find below the answers (in italics) to the short reviews prior to publication in ACPD. For technical reasons it has not been possible to integrate such changes in the ACPD manuscript. In agreement with the editorial office, the changes are published as comments and they will be integrated into the manuscript in the next iteration of revisions.

Anonymous Referee #1

Figure 2 is blurred, error bars and asterisks are not well visible.

Figure 4 could be considerably condensed using narrower bars. The line width of the error bars should be increased.

I recommend updating both figures prior to publication in ACPD.

Answer: *We thank the reviwer for the suggestions. Please find below the updated Figure 2 and Figure 4.*

[Figure]

**Figure 2**. Left: Emission of isoprene (a), MVK+MACR+ISOPOOH (b), products from MVK reduction (MEK (c) and 2-butanol (d)) and respiration (e) upon exposure to moderate (35°C) or severe (45°C) heat stress, and recovery to 25°C by red oak plants in the dark. Asterisks indicate significant differences (Kruskal-Wallis, $p<0.05$) compared to the unstressed

(25°C) case. Right: Emission of isoprene (f), MVK+MACR+ISOPOOH (g), products from MVK reduction (MEK (h) and 2-butanol (i)), and photosynthesis (j) upon exposure to moderate (35°C) or severe (45°C) heat stress, and recovery to 25°C in red oak plants in the light (ca. 90 µmol m$^{-2}$ s$^{-1}$ light intensity). Asterisks indicate significant differences (Kruskal-Wallis, $p < 0.05$) compared to the unstressed (25°C) control.

[Figure]

**Figure 4**. Emission of isoprene (a), MVK+MACR+ISOPOOH (b), MEK (c), and 2-butanol (d) in red oak plants heat stressed at 45°C for two hours in the light, under $^{12}CO_2$ or $^{13}CO_2$ atmosphere. Dashed horizontal lines represent natural abundances of isotopic compounds.

Anonymous Referee #2

The goal of the manuscript was to study the within-plant oxidation of isoprene to carbonyl compounds. Although isoprene emissions and its importance in plant stress reactions have been widely studied, its within-plant transformation to methacrolein, methylvinylketone and other VOCs has still been questionable. The study is well planned, accomplished and the MS is well written. There are no serious flaws, only some small mistakes. For example, I did not understand, how heat stress was applied or how was it possible to maintain a constant temperature? Yet, as the authors have published papers about isoprene emission earlier, I am sure that the description of the technical details is a little bit overlooked. In addition, I would add SD or SE values to the data of Figure 1 to show, that the results are based on several biological replicates.

I think the results are suitable for further revision and discussion.

Answer: *We thank the reviewer for the suggestions. Please find below the modification to the description and the updated Figure 1.*

*In section 2.1, page 8, line 14:*

Plants were watered every three days and developed fully expanded leaves after five weeks. Four days prior to the experiments, plants were transferred from the growth chamber into a climate cabinet (Climacell 707, BMT Medical Technology, Brno, Czech Republic). The climatic cabinet was employed to maintain constant climatic conditions (fumigation experiments) or to apply heat stress while maintaining all other parameters constant (heat stress experiments). The early transfer allowed plant recovery from accidental mechanical injures and adaptation to the new environment. The climate cabinet was set with the same parameters of the greenhouse, but at a constant temperature of 25°C (except for heat stress experiments), and interfaced with the PTR-ToF-MS via polyetheretherketone (PEEK) capillary tubes (ca. 1.5 m length x 1.01 mm ID, temperature: 110°C, flow: 40 sccm).

[Figure]

**Figure 1**. Uptake and transformation of MVK (a) and MACR (e) and of MVK transformation products, MEK (b), 2-butanol (c), 3-buten-e-ol (d) by red oak leaves. Results are reported as mean ± standard deviation (n=3). Negative values

denote uptake, while positive values indicate emission. Black arrows indicate the beginning and the end of the fumigation. In panel (b), the pulses of MEK uptake at the beginning of the fumigation and of MEK release after the end of the fumigation correspond to formation and release, respectively, of a MEK pool dissolved into the leaf water. At equilibrium the uptake of MEK and subsequent release of 2-butanol is constant and indicates a constant rate of MEK transformation within leaves. Analogous considerations can be made for 2-butanol and 3-buten-2-ol in panels (c) and (d), respectively.

---

## Referee Comment (RC1) · Anonymous Referee #1 · 3 Dec 2018

General comments:

Cappellin et al. describe a possible mechanism for the biogenic production of methyl ethyl ketone (MEK) from both exogenous and endogenous methyl vinyl ketone (MVK), which is decoupled from the plant's isoprene synthesis.

Earlier studies (even by one of the co-authors of this study) attributed isoprene an anti-oxidative role in plants, which was explained with its capability to capture reactive oxygen species (and thereby being oxidized to MVK/MACR). Over the last years, different studies have questioned this assumption. Cappellin et al. unequivocally show here that MVK production within plants under heat stress is not necessarily linked to the

plant's capability to synthesize isoprene. Therefore, the manuscript has the potential to become an important contribution to the controversial discussion, whether isoprene exerts an antioxidant role in plants.

In general, the manuscript is very well written, in a clear and concise way. However, I'm struggling a bit with the experimental design and the data interpretation. The Methods part misses details on peak assignment in PTR/SRI-ToF-MS, and $CO_2$ measurements/calibrations (see comments below). The number of replicates (3) in each experiment is borderline. This is also reflected in the large error bars in Figure 2.

It is pretty brave to make statements on interconversion of in part isomeric compound using solely PTR-MS. Even when using NO+ ions for chemical ionization in the PTR/SRI-ToF-MS you have a lot of interfering ions from the compounds you were investigating. Moreover, natural isotopes of some of the investigated compounds could interfere with the parent ions of other compounds. The description of the data analysis in the methods section does not reveal if this effect was taken into account, nor does it explain satisfactorily how the ions signals were attributed to the different compounds. Especially in the case of the various alcohols a proper identification seems almost impossible with the instrumentation you used. I would expect to have a table containing all the different compounds and the associated ions in the two measurement modes of the PTR/SRI-ToF-MS. This would allow the reader to better judge whether the peak assignment is justified.

To my mind, such an experiment would have strongly benefited from additional analyses capable to distinguish isomeric compounds, such as GC-MS or similar.

Although I know it is a lot of work, I would recommend to perform additional experiments and to trap VOCs for GC-MS analyses in order to eliminate any doubt in the interpretation of the data.

The quality of some of the original figures was very bad (the updated ones submitted as Author Comment are OK).

Specific comments:

p. 2, line 18: was there really formed any 3-buten-2-ol? In Figure 1a, the 3-buten-2-ol seems to be zero throughout the whole experiment.

p. 2, lines 21-22: you disregard here that there is no possible direct conversion of 3-buten-2-ol to MEK. How sure are you about these data? I guess it is really tricky to properly distinguish 3-buten-2-ol from MEK using solely PTR-MS.

p. 4, lines 13-15: The calculated assimilation rate is very low. I've never seen assimilation rates in a comparable range as the dark respiration values in a light (!) experiment. What was the PAR you used in these experiments? Apparently, you used the PTR-ToF-MS to measure $CO_2$ levels: have you considered different humidities in dark/light experiments when calibrating the PTR-ToF-MS for $CO_2$? Can you comment on the accuracy of this method to measure $CO_2$?

p. 5, lines 20-23: These reaction yield calculations require further explanations, either here or in the Methods section. Where do you "SHOW" that 73% of MVK is converted into MEK?

p. 8, line 16: is there a reason why you heated your sample line to 110°C? At such high temperatures you may encounter surface assisted reactions and thermal decomposition of larger compounds, possibly interfering with the ion signals of interest. The compounds you were interested in should all be fairly volatile, excessive line heating is therefore counter-productive in this case.

p. 8, line 27: I guess this is 10ul/min of liquid standard. What is the actual volume mixing ratio of the compound in the VOC-bag inlet air?

p. 9, line 3: what were the $CO_2$ concentrations at the outlet of your VOC-bag? Depending on the enclosed leaf area, during light conditions at this modest flow rates you might have run into $CO_2$ deficit conditions for your plant. This could have affected your measured VOC signals.

Figure 1: are these data of a single experiment or the mean over several experiments? This should be indicated in the figure caption. Since you have a possible interconversion of the measured compounds as well as emission and re-uptake, the y-axes should be labeled "Net VOC flux".

Figure 2: The overall quality of this figure is very bad! The resolution is indisputably low. The error bars and asterisks are almost not visible. I assume the compound grouping here is based on the different ion signals when using H3O+ ions for chemical ionization in PTR-MS, yielding the same ion for the different groups. This should be stated somewhere. As you are focusing on endogenously formed compounds here, it would make sense to normalize the signals measured in the different conditions to the stomatal conductance of the leaves. This way you might get an idea on the actual concentration of these compounds within the leaves.

Figure 5: you completely neglect the conversion of 3-buten-2-ol to MEK here, although, considering Figure 1, this seems its major conversion pathway. Again, the resolution of the background image could be improved.

Technical corrections:

p. 2, line 3: remove "plenty of". How can you claim there is plenty of evidence for the heat dissipating and thylakoid membranes stabilizing properties of isoprene, when you cite only two publications? Btw: how large can the heat dissipating effect of isoprene be when you compare the isoprene emission fluxes ($nmol/(m^2{*}s)$) with leaf transpiration ($mmol/(m^2{*}s)$)?

p. 2, lines 19-20: remove this sentence. Why would the plant produce isoprene to scavenge ROS, if the isoprene oxidation products are similarly cytotoxic and in turn need to be scavenged themselves?

p. 3, lines 2: "..., though the full mechanism was not described.": nor is it described here. What are the enzymes involved in the detoxification reactions? Just saying.

p. 4, line 14: this is no proper sentence. You compare assimilation and isoprene emission values with a light intensity.

p. 5, line 18: Fig. 5 is a possible pathway for the biogenic formation and emission of MEK, but does not really explain it. A proper explanation would require the investigation of the enzymatic pathways involved in the MEK production.

p. 5, line 19: The results suggest that WITHIN PLANT isoprene oxidation is not the source of these VOCs IN YOUR EXPERIMENT! Atmospheric oxidation of isoprene is undoubtedly the main source of MVK in the atmosphere.

p. 10, lines 4-5: the cited reference does not contain any information on the spectral peaks to monitor!

---

## Referee Comment (RC2) · Anonymous Referee #2 · 5 Dec 2018

Generally, I think that the study offers new knowledge in the area of transformation pathways of carbonyl compounds to 2-butanol and 3-buten-2-ol. The results are valuable and will inspire researchers to test the new hypothesis. For example, in this study with red oak, the authors did find a link between isoprene and methylvinylketone. Yet, I wonder, whether the link could be characteristic to some other plant species? In similar as in 2012 Jardine et al published a correlation between the emissions of isoprene and methacrolein, but in the present study, that correlation was missing. By the way, although testing of methacrolein did not give expected results, I still recommend adding its molecular structure to Figure 5. I find that the study is done by using suitable analytical methods and the conclusions are all appropriate. MS is well written and there

are no serious flaws. According to the initial MS evaluation, the authors have improved the figures. The MS is well structured and the abstract provides a complete summary of the results.

Minor criticism In the Abstract the abbreviation of MVK is not explained

In the Abstract L 7, please change ′. . . we show that MVK and MEK emission caused by heat stress is. . .′ to ′. . . we show that MVK and MEK emissions caused by heat stress are. . .′

Among Keywords and later in the text (P 3 L9 and 22) please change ′2-butenol′ to ′2-butanol′

At the beginning of the Introduction P1, I would change ′has′ to ′have′ in ′. . .and particularly the latest studies employing genetic engineering to produce transgenic plants with modified isoprene emission, has . . .′ P 2 L3 change ′there is plenty of evidence that isoprene positively influence..′ to ′there is plenty of evidence that isoprene positively influences..′

At the beginning of Results L4 ′ MVK reduction generated either MEK or 3-buten-2-ol,′ - according to Figure 2 I would say 2-butanol instead of 3-buten-2-ol

At the beginning of P4 starting from L2 ′Emission of MVK+MACR+ISOPOOH was detected at a level of $0.001\pm0.002$ nmol m-2s-1′ - by comparing the text to the figure it seems that the number should be multiplied with 1000, no? The same mistake is repeated in the following sentences.

Abbreviation of methylvinylketone, methylethylketone and ROS should be explained again at the beginning of the Discussion

P5 L 7 to 9 I would use the plural instead of singular ′Emission of MVK and MEK has been rarely measured at leaf level and is challenging to measure because i) emission is small, especially in the case of MVK; ii) separation of leaf and atmospheric sources in the 10 presence of high isoprene emissions is difficult. Jardine and co-workers (Jardine

et al., 2012, 2013) reported emission of methacrolein (MACR) and methyl vinyl ketone (MVK) in isoprene emitting trees. ′ to ′Emissions of MVK and MEK have been rarely measured at leaf level and are challenging to measure because i) emissions are small, especially in the case of MVK; ii) separation of leaf and atmospheric sources in the presence of high isoprene emissions is difficult. Jardine and co-workers (Jardine et al., 2012, 2013) reported the emissions of methacrolein (MACR) and methyl vinyl ketone (MVK) in isoprene emitting trees. ′

P9 L9 I would change ′...repeated switching..′ to ′repeated by switching ...′

Figure 1 - please add titles to y-axes, change the unit to ′nmol m-2 s-1′ and it is sufficient to write ′Time (min)′ only under panel e

Also, I would change ′Uptake and transformation of MVK (a) and MACR (e) and of MVK transformation products, MEK (b), 2-butanol (c), 3-buten-e-ol (d) by red oak leaves. Negative values denote uptake, while positive values indicate emission. ′ as follows ′Uptake of MVK (a), MEK (b), 2-butanol (c), 3-buten-e-ol (d) and MACR (e) and emission of their oxidized or reduced products in red oak leaves. Negative values denote the uptake - and positive values the emission of volatiles. ′

Please check also the rest of the figure legends.

Figure 2 – Is it possible to show or mark in the text the proportion of methacrolein (MACR) in the mixture of MVK+MACR+ISOPOOH? Does the figure show mean $\pm$ SE or mean $\pm$ SD? I would delete negative error bars. Please use ′min′ instead of ′minutes′.

---

## Author Response (AR1)

Dear Editor,

We thank the two reviewers for their thoughtful and detailed comments on our manuscript. Please find below our response to the comments raised by the reviewers. We hope to have satisfactorily responded to all reviewer suggestions and comments, which have substantially improved the manuscript. Changes in the revised manuscript were made using the track-mode tool and are visible. Author comments below are in bold.

**REVIEWER #2**

Generally, I think that the study offers new knowledge in the area of transformation pathways of carbonyl compounds to 2-butanol and 3-buten-2-ol. The results are valuable and will inspire researchers to test the new hypothesis. For example, in this study with red oak, the authors did find a link between isoprene and methylvinylketone. Yet, I wonder, whether the link could be characteristic to some other plant species? In similar as in 2012 Jardine et al published a correlation between the emissions of isoprene and methacrolein, but in the present study, that correlation was missing. By the way, although testing of methacrolein did not give expected results, I still recommend adding its molecular structure to Figure 5.

**We thank the reviewer for these comments. We tested the MVK transformation mechanism on two other plant species beside red oak (i.e. *Hedera helix and Vitis vinifera*) using GC-MS and did find analogous results to red oak. Such data is reported in the Supplementary Information of the revised manuscript. Since Figure 5 refers specifically to within-plant MVK transformation, we suggest not to add methacrolein. In the supplement, Figure S1 was added.**

[Figure]

**Figure S1.** Excerpt of qualitative GC-MS analysis of six untreated plants (upper figure) and of the same plants fumigated with MVK (lower figure, n=6). The reported chromatograms correspond to m/z = 45 Th, so that all relevant peaks are clearly visible. Different colours indicate different plants (yellow for *Hedera helix*; black for *Vitis vinifera*; other colours for *Quercus rubra*). Upon MVK fumigation, MEK and 2-butanol were formed by all plants, while 3-buten-2-ol was below detection limit and could only be detected by PTR-ToF-MS (Table S1). No other MVK transformation compounds were detected by GC-MS. Such data is reported to support compound identification for the fumigation experiments reported in the main text (Figure 1).

**In the supplement, the following paragraphs were added on page 1, line 1-30:**

**S.1 Complementary GC-MS analysis of MVK transformation products**

In order to support compound identification of MVK transformation products by red oak plants, the MVK fumigation experiments described in the main text were repeated with some minor modifications to include qualitative analysis by solid phase microextraction - gas chromatography – mass spectrometry (SPME-GC-MS). As depicted in Figure S1, MEK and 2-butanol were unambiguously identified as MVK transformation products by red oaks, confirming the results obtained using PTR/SRI-ToF-MS (Figure 1 of the main text). Similar results were obtained for *Hedera helix* and *Vitis vinifera* (Figure S1). No other putative MVK transformation products were detected by the SPME-GC-MS analysis. In particular, 3-buten-2-ol was below detection limit. At the contrary, the corresponding PTR-ToF-MS analysis reported in Figure 1 and Table S1, besides MEK and 2-butanol, also detected a small but statistically significant emission of 3-buten-2-ol (Table S1). The identification of the PTR-ToF-MS signal corresponding to the ion $C_4H_7^+$ as 3-buten-2-ol, besides being consistent with measurements using the pure compound standard (3-buten-2-ol undergoes protonation followed by dehydration upon reaction with $H_3O^+$), has also theoretical reasons. Ketones have been reported to be transformed by plants via reduction reactions (Kergomard et al., 1988). Hence MEK and 3-buten-2-ol are expected as MVK transformation products. Moreover, reduction reactions occur for MACR, producing in particular isobutyraldehyde and 2-methallyl alcohol (Figure 1, Table S1, and Muramoto *et al.* (2015)).

**S.1.1 Experimental setup for SPME-GC-MS analysis**

The plant management and experimental setup was analogous to the one described in main text with the addition of a SPME fiber in the VOC-bag for collecting VOC for subsequent GC-MS analysis. No PTR/SRI-ToF-MS measurements were performed in this case. The experiment was repeated on six plants (four *Quercus rubra*, one *Hedera helix*, one *Vitis vinifera*). The GC-MS analysis was carried out as follows.
Headspace volatile compounds were collected by a 2 cm Solid Phase Microextration fibre coated with divinylbenzene/carboxen/polydimethylsiloxane 50/30 lm (DBV/CAR/PDMS, Sigma-Adrich, St. Lewis, USA), inserted through the VOC-bag using a manual holder. The fibre was exposed to the headspace air through the duration of the fumigation. Volatile compounds adsorbed on the SPME fibre were desorbed at 250°C in the injector port of a GC interfaced with a mass detector (GC Agilent 7820A with Agilent 5977B MSD, Agilent Technologies, Santa Clara CA, USA). The mass detector was operated in electron ionization mode (EI, internal ionization source; 70 eV) with scan range from m/z 25–200. Separation was achieved on a Supelco SPB-624 capillary column (20 m x 0.18 mm ID x 1 µm film thickness; Sigma-Adrich, St. Lewis, USA). The GC oven temperature program consisted in 40°C for 6 min, then 40–200°C at 40°C min[-1], and stable at 200°C for 5 min. Helium was used as the carrier gas with a constant column flow rate of 0.8 mL min[-1]. Compound identification was based on mass spectra matching with the standard NIST libraries (NIST 2.2 2014) and retention times of authentic reference standards.

I find that the study is done by using suitable analytical methods and the conclusions are all appropriate. MS is well written and there are no serious flaws. According to the initial MS evaluation, the authors have improved the figures. The MS is well structured and the abstract provides a complete summary of the results.
**We thank the reviewer for these comments.**

Minor criticism

In the Abstract the abbreviation of MVK is not explained
**This point has been corrected**

In the Abstract L 7, please change '…we show that MVK and MEK emission caused by heat stress is…' to '…we show that MVK and MEK emissions caused by heat stress are…'
**This point has been corrected.**

Among Keywords and later in the text (P 3 L9 and 22) please change '2-butenol' to '2-butanol'
**This point has been corrected.**

5  At the beginning of the Introduction P1, I would change 'has' to 'have' in '…and particularly the latest studies employing genetic engineering to produce transgenic plants with modified isoprene emission, has…' P 2 L3 change 'there is plenty of evidence that isoprene positively influence… to 'there is plenty of evidence that isoprene positively influences…'
**This point has been corrected.**

At the beginning of Results L4 'MVK reduction generated either MEK or 3-buten-2-ol,'
- according to Figure 2 I would say 2-butanol instead of 3-buten-2-ol
**It is true that the main final transformation products are either MEK or 2-butanol, as shown in the figure. However, in that sentence we are referring to the compounds generated upon**
15  **reduction of the either the alkene moiety of MVK, leading to MEK, or the carbonyl moiety, leading to 3-buten-2-ol. We suggest not to change that sentence.**

At the beginning of P4 starting from L2 'Emission of MVK+MACR+ISOPOOH was detected at a level of 0.001±0.002 nmol m-2s-1'- by comparing the text to the figure
20  it seems that the number should be multiplied with 1000, no? The same mistake is repeated in the following sentences.
**A factor of $10^{-3}$ is reported at the top of the axis, this may have generated the confusion. We revised the figure (converting the y axis in pmol $m^{-2}s^{-1}$) in order to make it clearer.**

25  Abbreviation of methylvinylketone, methylethylketone and ROS should be explained again at the beginning of the Discussion
**This point has been corrected.**

P5 L 7 to 9 I would use the plural instead of singular 'Emission of MVK and MEK has
30  been rarely measured at leaf level and is challenging to measure because i) emission is small, especially in the case of MVK; ii) separation of leaf and atmospheric sources in the 10 presence of high isoprene emissions is difficult. Jardine and co-workers (Jardine et al., 2012, 2013) reported emission of methacrolein (MACR) and methyl vinyl ketone (MVK) in isoprene emitting trees.' to 'Emissions of MVK and MEK have been rarely
35  measured at leaf level and are challenging to measure because i) emissions are small, especially in the case of MVK; ii) separation of leaf and atmospheric sources in the presence of high isoprene emissions is difficult. Jardine and co-workers (Jardine et al., 2012, 2013) reported the emissions of methacrolein (MACR) and methyl vinyl ketone (MVK) in isoprene emitting trees. '
40  **We agree with the reviewer. This point has been corrected.**

P9 L9 I would change '…repeated switching…' to 'repeated by switching'
**This point has been corrected.**

Figure 1 - please add titles to y-axes, change the unit to 'nmol m-2 s-1' and it is sufficient to write
'Time (min)' only under panel e
**This point has been corrected**

Also, I would change 'Uptake and transformation of MVK (a) and MACR (e) and of MVK
transformation products, MEK (b), 2-butanol (c), 3-buten-e-ol (d) by red oak leaves.
Negative values denote uptake, while positive values indicate emission. ' as follows
'Uptake of MVK (a), MEK (b), 2-butanol (c), 3-buten-e-ol (d) and MACR (e) and emission of their
oxidized or reduced products in red oak leaves. Negative values denote
the uptake - and positive values the emission of volatiles. '
**We agree with the reviewer. This point has been corrected.**

Please check also the rest of the figure legends
**We checked the figure legends.**

Figure 2 – Is it possible to show or mark in the text the proportion of methacrolein (MACR) in the
mixture of MVK+MACR+ISOPOOH? Does the figure show mean ± SE or mean ± SD? I would delete
negative error bars. Please use 'min' instead of 'minutes'.
**It is not possible to mark the proportion of methacrolein as these data correspond to
measurement using H3O+ as primary ions, therefore only the sum of MVK+MACR+ISOPOOH
is measured. In the caption of figure 2, we clarified that "Results are reported as mean ± standard
error (n=6)". In the Figure 2, we replaced "minutes" with "min". We delete negative error bars as
suggested.**

**REVIEWER #1**

General comments: Cappellin et al. describe a possible mechanism for the biogenic production of
methyl ethyl ketone (MEK) from both exogenous and endogenous methyl vinyl ketone (MVK), which
is decoupled from the plant's isoprene synthesis.

Earlier studies (even by one of the co-authors of this study) attributed isoprene an anti-oxidative role in
plants, which was explained with its capability to capture reactive oxygen species (and thereby being
oxidized to MVK/MACR). Over the last years, different studies have questioned this assumption.
Cappellin et al. unequivocally show here that MVK production within plants under heat stress is not
necessarily linked to the plant's capability to synthesize isoprene. Therefore, the manuscript has the
potential to become an important contribution to the controversial discussion, whether isoprene exerts
an antioxidant role in plants.
**We thank the reviewer for these comments.**

In general, the manuscript is very well written, in a clear and concise way. However, I'm struggling a bit with the experimental design and the data interpretation. The Methods part misses details on peak assignment in PTR/SRI-ToF-MS, and CO2 measurements/calibrations (see comments below). The number of replicates (3) in each experiment is borderline. This is also reflected in the large error bars in
5 Figure 2.
**We added details on peak assignment in PTR/SRI-ToF-MS and CO₂ measurements/calibrations (see answer to comments below). We revised Figure 2 using 6 replicates as we already had the data (see answer to comments below). For the transformation mechanism we added GC-MS data for 6 plants (see answer to comments below). The statistical significance of the conclusions is solid.**

It is pretty brave to make statements on interconversion of in part isomeric compound using solely PTR-MS. Even when using NO+ ions for chemical ionization in the PTR/SRI-ToF-MS you have a lot of interfering ions from the compounds you were investigating. Moreover, natural isotopes of some of the investigated compounds could interfere with the parent ions of other compounds. The description of the
15 data analysis in the methods section does not reveal if this effect was taken into account, nor does it explain satisfactorily how the ions signals were attributed to the different compounds. Especially in the case of the various alcohols a proper identification seems almost impossible with the instrumentation you used. I would expect to have a table containing all the different compounds and the associated ions in the two measurement modes of the PTR/SRI-ToF-MS. This would allow the reader to better judge
20 whether the peak assignment is justified.
**We added a table containing all the different compounds and the associated ions in the two measurement modes of the PTR/SRI-ToF-MS in the Supplementary Information.**
**In the supplement, the following table was added: Table S2.**

25 Table S2. Spectral peaks and corresponding ions in PTR/SRI-TOF-MS for $H_3O^+$ mode and $NO^+$ mode. The mass resolving power of the PTR/SRI-ToF-MS (> 4000) and the peak deconvolution algorithm used (Cappellin et al., 2011) allowed to resolve the peaks reported in this table in each mode.

| Compound | H3O+ mode | | NO+ mode | |
| --- | --- | --- | --- | --- |
| | *m/z* | Ion sum formula | *m/z* | Ion sum formula |
| Isoprene | | | | |
| | 69.07 | $C_5H_8H^+$ | 68.062 | $C_5H_8^+$ |
| | 70.0732 | $^{13}CC_4H_8H^+$ | 69.0654 | $^{13}CC_4H_8^+$ |
| | 71.0766 | $^{13}C_2C_3H_8H^+$ | 70.069 | $^{13}C_2C_3H_8^+$ |
| | 72.0844 | $^{13}C_3C_2H_8H^+$ | 71.0721 | $^{13}C_3C_2H_8^+$ |
| | 73.0833 | $^{13}C_4CH_8H^+$ | 72.0754 | $^{13}C_4CH_8^+$ |
| | 74.0866 | $^{13}C_5H_8H^+$ | 73.0788 | $^{13}C_5H_8^+$ |
| | | | 98.06 | $C_5H_8 \cdot NO^+$ |
| | | | 99.0634 | $^{13}CC_4H_8 \cdot NO^+$ |
| | | | 100.0667 | $^{13}C_2C_3H_8 \cdot NO^+$ |
| | | | 101.0701 | $^{13}C_3C_2H_8 \cdot NO^+$ |
| | | | 102.0734 | $^{13}C_4CH_8 \cdot NO^+$ |
| | | | 103.0768 | $^{13}C_5H_8 \cdot NO^+$ |
| MVK | | | | |
| | 71.0491 | $C_4H_6OH^+$ | 100.0393 | $C_4H_6O \cdot NO^+$ |
| | 72.0525 | $^{13}CC_3H_6OH^+$ | 101.0427 | $^{13}CC_3H_6O \cdot NO^+$ |
| | 73.0558 | $^{13}C_2C_2H_6OH^+$ | 102.046 | $^{13}C_2C_2H_6O \cdot NO^+$ |
| | 74.0592 | $^{13}C_3CH_6OH^+$ | 103.0494 | $^{13}C_3CH_6O \cdot NO^+$ |
| | 75.0626 | $^{13}C_4H_6OH^+$ | 104.0527 | $^{13}C_4H_6O \cdot NO^+$ |
| MEK | | | | |
| | 73.0648 | $C_4H_8OH^+$ | 102.05495 | $C4H8O \cdot NO+$ |

| | 74.0681 | $^{13}CC_3H_8OH^+$ | 103.0583 | $^{13}CC3H8O\cdot NO+$ |
|---|---|---|---|---|
| | 75.0715 | $^{13}C_2C_2H_8OH^+$ | 104.0627 | $^{13}C2C2H8O\cdot NO+$ |
| | 76.0748 | $^{13}C_3CH_8OH^+$ | 105.065 | $^{13}C3CH8O\cdot NO+$ |
| | 77.0782 | $^{13}C_4H_8OH^+$ | 106.0684 | $^{13}C4H8O\cdot NO+$ |
| 3-buten-2-ol | | | | |
| | 55.0542 | $C_4H_7^+$ | 71.0491 | $C_4H_7O^+$ |
| | 56.0576 | $^{13}CC_3H_7^+$ | 72.0525 | $^{13}CC_3H_7O^+$ |
| | 57.0609 | $^{13}C_2C_2H_7^+$ | 73.0558 | $^{13}C_2C_2H_7O^+$ |
| | 58.0643 | $^{13}C_3CH_7^+$ | 74.0592 | $^{13}C_3CH_7O^+$ |
| | 59.0676 | $^{13}C_4H_7^+$ | 75.0626 | $^{13}C_4H_7O^+$ |
| 2-butanol | | | | |
| | 57.07 | $C_4H_9^+$ | 73.0648 | $C_4H_9O^+$ |
| | 58.0732 | $^{13}CC_3H_9^+$ | 74.0681 | $^{13}CCH_9O^+$ |
| | 59.0766 | $^{13}C_2C_2H_9^+$ | 75.0715 | $^{13}C_2C_2H_9O^+$ |
| | 60.0799 | $^{13}C_3CH_9^+$ | 76.0748 | $^{13}C_3CH_9O^+$ |
| | 61.0833 | $^{13}C_4H_9^+$ | 77.0782 | $^{13}C_4H_9O^+$ |

To my mind, such an experiment would have strongly benefited from additional analyses capable to distinguish isomeric compounds, such as GC-MS or similar.

5  Although I know it is a lot of work, I would recommend to perform additional experiments and to trap VOCs for GC-MS analyses in order to eliminate any doubt in the interpretation of the data.
**We agree with the reviewer. Therefore, we carried out additional experiments using GC-MS. In the Supplementary Information we added the GC-MS data to support compound identification. Most importantly, methyl ethyl ketone and 2-butanol were unambiguously identified as**

10  **transformation products of methyl vinyl ketone using both comparison to the NIST database of EI spectra and retention time of pure standards. 3-buten2-ol is a very minor product and was below detection limit in GC-MS. Only the high sensitivity of the PTR-TOF allowed detecting it. Its identification as 3-buten-2-ol among the possible isomers has theoretical reasons since ketones have been reported to be transformed by plants via reduction reactions (Kergomard et al., 1988),**

15  **suggesting that MEK and 3-buten-2-ol are expected as MVK transformation products. No other transformation products were detected by GC-MS. The compound identification for the transformation products of methacrolein is supported by GC-MS data present within the literature (Muramoto *et al.*, 2015). In conclusion the additional GC-MS agree with the compound identification reported in the manuscript. We thank the reviewer for suggesting this**

20  **improvement.**
**We added Sections S.1 and S.1.1 and Figure S1 in the Supplementary Information. They are also reported in an answer to the other reviewer.**

The quality of some of the original figures was very bad (the updated ones submitted
25  as Author Comment are OK).

Specific comments:

p. 2, line 18: was there really formed any 3-buten-2-ol? In Figure 1a, the 3-buten-2-ol
30  seems to be zero throughout the whole experiment.

**Yes, 3-buten-2-ol was formed as a minor product. Since this was not evident in Figure 1a, Table S1 was added in the Supplementary Information.**
**In the Supplement Table S1 was added.**

Table S1. Total net flux in the fumigation experiments reported in Figure 1 (main text). Positive values denote net emission while negative values denote net uptake. Results are obtained by integrating the net flux curves reported in Figure 1. Results are reported as *mean ± se*.

| *Fumigation with MVK* | TOTAL NET FLUX ($\mu gC/m^2$) | % of MVK uptake |
|---|---|---|
| MVK | -836 ± 257 | |
| MEK | 659 ± 242 | 73% ± 6% |
| 3-buten-2-ol | 38 ± 13 | 4% ± 0.4% |
| 2-butanol | 207 ± 32 | 19% ± 2% |
| MEK + 3-buten-2-ol + 2-butanol | 839 ± 281 | 97% ± 4% |
| *Fumigation with MEK* | TOTAL NET FLUX ($\mu gC/m^2$) | % of MEK uptake |
| MVK | n.d. | |
| MEK | -270 ± 53 | |
| 3-buten-2-ol | n.d | |
| 2-butanol | 171 ± 34 | 87% ± 13% |
| *Fumigation with 2-butanol* | TOTAL NET FLUX ($\mu gC/m^2$) | % of 2-butanol uptake |
| MVK | n.d. | |
| MEK | 47 ± 6 | 90% ± 29% |
| 3-buten-2-ol | n.d | |
| 2-butanol | -63 ± 10 | |
| *Fumigation with 3-buten-2-ol* | TOTAL NET FLUX ($\mu gC/m^2$) | % of 3-buten-2-ol uptake |
| MVK | n.d. | |
| MEK | 33 ± 7 | 33% ± 10% |
| 3-buten-2-ol | -107 ± 10 | |
| 2-butanol | 14 ± 2 | 14% ± 4% |
| MEK + 2-butanol | 48 ± 9 | 47% ± 14% |
| *Fumigation with MACR* | TOTAL NET FLUX ($\mu gC/m^2$) | % of MACR uptake |
| MACR | -431 ± 110 | |
| Isobutyraldehyde | 33 ± 9 | 6.5% ± 0.8% (6.4%*) |
| 2-Methallyl alcohol | 4 ± 1 | 0.7% ± 0.3% (1.1%*) |
| Isobutanol | 10 ± 1 | 2.1% ± 0.3% (1.8%*) |
| Isobutyraldehyde + 2-Methallyl alcohol + Isobutanol | 48 ± 11 | 9.3% ± 0.8% |

*\* Results in brackets represents values reported by Muramoto et al. (2015) in similar experiments using tomato plants.*

p. 2, lines 21-22: you disregard here that there is no possible direct conversion of 3-buten-2-ol to MEK. How sure are you about these data? I guess it is really tricky to properly distinguish 3-buten-2-ol from MEK using solely PTR-MS.
**It is possible to distinguish 3-buten-2-ol from MEK using PTR-TOF. In fact, in $H_3O^+$ mode MEK undergoes proton transfer leading to $C_4H_9O^+$ at 73.0648 Th, while 3-buten-2-ol also undergoes dehydration leading to $C_4H_7^+$ at 55.0542 Th. We agree with the reviewer that the conversion of 3-buten-2-ol to MEK should be added.**
**In page 3, line 22, the following sentence was added:**

**"Conversion of 3-buten-2-ol to MEK is also possible (Figure 1d)."**

p. 4, lines 13-15: The calculated assimilation rate is very low. I've never seen assimilation rates in a comparable range as the dark respiration values in a light (!) experiment.

What was the PAR you used in these experiments? Apparently, you used the PTRToF-MS to measure $CO_2$ levels: have you considered different humidities in dark/light experiments when calibrating the PTR-ToF-MS for CO2? Can you comment on the accuracy of this method to measure CO2?

**We reported the calibration curve and the correction factors used at different humidities in the Supplementary Material (Figure S2). The method is estimated to have an accuracy of about 5% using an integration time of 90 s and its applicability is limited to CO2 concentrations in the ppm range since the sensitivity is very much lower than typical PTR-MS sensitivities for VOCs in general. It has been used in this case since a CO2 detector was not available. As reported in the p.8 line 12, the PAR was 90 $\mu$mol m$^{-2}$ s$^{-1}$. The results for both the assimilation rates and the isoprene emission are comparable with those of previous literature experiments on the same plant species (Loreto and Sharkey, 1990). The results for dark respiration are also comparable with the values reported by other studies on red oak (Turnbull et al., 2001).**
**In the Supplement, Figure S2 was added.**

p. 5, lines 20-23: These reaction yield calculations require further explanations, either here or in the Methods section. Where do you "SHOW" that 73% of MVK is converted into MEK?

**We added a table reporting all conversion percentages (including MVK to MEK).**
**In the Supplementary Information, we added Table S1.**

p. 8, line 16: is there a reason why you heated your sample line to 110∘C? At such high temperatures you may encounter surface assisted reactions and thermal decomposition of larger compounds, possibly interfering with the ion signals of interest. The compounds you were interested in should all be fairly volatile, excessive line heating is therefore counter-productive in this case.

**We clarified that only the last part (about 1 m) of the PEEK sampling line was heated. It avoids memory effects and condensation. The residence time is low (PEEK inner diameter 1.01 mm, 40 sccm flow) and no evidence of surface assisted reactions and thermal decomposition of larger compounds interfering with the ion signal of interest was found. Moreover, our conclusions are based on the simultaneous behavior of several compounds (e.g. MEK, 3-butenl-2-ol, 2-butanol). It is highly unlikely to have interference in all channels.**
**At page 8, line 16, the sentence now reads:**
**"(…) interfaced with the PTR-ToF-MS via polyetheretherketone (PEEK) capillary tubes (ca. 1.5 m length x 1.01 mm ID, temperature: 110°C (part outside the climatic chamber, ca. 1 m), flow: 40 sccm)".**

p. 8, line 27: I guess this is 10ul/min of liquid standard. What is the actual volume

mixing ratio of the compound in the VOC-bag inlet air?

**We clarified that the actual volume mixing ratios of the compounds in the VOC-bag inlets were the following: 290 ppbv for MVK, 330 ppbv for MEK, 90 ppbv for 2-butanol, 96 ppbv for 3-buten-2-ol, 140 ppbv for MACR.**

5 **At page 8, line 16, the following sentence was added:**
**"The volume mixing ratio of the compound used for the fumigation in the VOC-bag inlets was the following: 290 ppbv for MVK, 330 ppbv for MEK, 90 ppbv for 2-butanol, 96 ppbv for 3-buten-2-ol, 140 ppbv for MACR".**

10 p. 9, line 3: what were the CO2 concentrations at the outlet of your VOC-bag? Depending on the enclosed leaf area, during light conditions at this modest flow rates you
might have run into CO2 deficit conditions for your plant. This could have affected your measured VOC signals.

**The $CO_2$ concentration (or $^{13}CO_2$ concentration) at the outlet of the VOC-bag was above 350 ppm**
15 **in all conditions, indicating that the plants did not experience carbon dioxide deficit conditions.**

Figure 1: are these data of a single experiment or the mean over several experiments?
This should be indicated in the figure caption. Since you have a possible interconversion of the measured compounds as well as emission and re-uptake, the y-axes should
20 be labeled "Net VOC flux".

**This figure was revised as part of the previous authors' comment. Figure 1 reports the mean over experiments. This point was clarified in the updated figure caption.**
**In updated Figure 1:**
**The y-axes were relabeled "Net VOC flux (nmol m$^{-2}$ s$^{-1}$)".**

Figure 2: The overall quality of this figure is very bad! The resolution is indisputably low. The error bars and asterisks are almost not visible. I assume the compound grouping here is based on the different ion signals when using H3O+ ions for chemical ionization in PTR-MS, yielding the same ion for the different groups. This should be stated somewhere. As you are focusing on endogenously formed
30 compounds here, it would make sense to normalize the signals measured in the different conditions to the stomatal conductance of the leaves. This way you might get an idea on the actual concentration of these compounds within the leaves.

**This figure was revised as part of the previous authors' comment. As the reviewer pointed out above, the quality of the updated figures is ok. Although we agree that in general it would be**
35 **interesting to get an idea of the actual compound concentration within the leaves, we propose not to carry out a further normalization since in the rational of the paper the figure is reported in order to prove that the compounds of interest are emitted under heat stress conditions. Such conclusion can be based on the compound net flux as reported in the updated figure.**
**We clarified that the compound grouping was based on H3O+ ionization in the caption of the**
40 **updated figure 2.**
**Figure 2 was revised to include six replicates and the values reported in page 3 line 29 – page 4 line 25 referring to Figure 2 were changed accordingly.**

Figure 5: you completely neglect the conversion of 3-buten-2-ol to MEK here, although, considering Figure 1, this seems its major conversion pathway. Again, the resolution of the background image could be improved.

5 **We added the arrow for conversion from 3-buten-2-ol to MEK, which was missing. We thank the reviewer for pointing this out. We improved the resolution of the background figure.**

Technical corrections:

10 p. 2, line 3: remove "plenty of". How can you claim there is plenty of evidence for the heat dissipating and thylakoid membranes stabilizing properties of isoprene, when you cite only two publications? Btw: how large can the heat dissipating effect of isoprene be when you compare the isoprene emission fluxes (nmol/(m2 *s)) with leaf transpiration (mmol/(m2 *s))?
**This point has been corrected.**
15 **At page 2, line 3:**
**We removed "plenty of".**

p. 2, lines 19-20: remove this sentence. Why would the plant produce isoprene to scavenge ROS, if the isoprene oxidation products are similarly cytotoxic and in turn need to be scavenged themselves?
20 **The sentence was removed.**

p. 3, lines 2: "..., though the full mechanism was not described.": nor is it described here. What are the enzymes involved in the detoxification reactions? Just saying.
**This point has been corrected.**
25 **At page 3, line 2, the sentence now reads:**
**"We suggested that MVK could be efficiently detoxified by reduction reactions which lead mostly to MEK and, to a minor extent to 2-butanol, though the full interconversion scheme was not studied."**

30 p. 4, line 14: this is no proper sentence. You compare assimilation and isoprene emission values with a light intensity.
**We clarified that we compared carbon assimilation and isoprene emission to literature values at analogous light intensity and the results were consistent.**
**At page 4, line 14, the sentence now reads:**
35 **"These values for carbon assimilation and isoprene emission are consistent literature findings, considering the light intensity to which plants were exposed (e.g. see (Loreto and Sharkey, 1990))."**

p. 5, line 18: Fig. 5 is a possible pathway for the biogenic formation and emission of MEK, but does not
40 really explain it. A proper explanation would require the investigation of the enzymatic pathways involved in the MEK production.
**This point has been corrected.**

**At page 5, line 18, the sentence now reads:**
**"This mechanism is the first possible pathway for the biogenic production and emission of MEK."**

p. 5, line 19: The results suggest that WITHIN PLANT isoprene oxidation is not the source of these VOCs IN YOUR EXPERIMENT! Atmospheric oxidation of isoprene is undoubtedly the main source of MVK in the atmosphere.
**This point has been corrected.**
**At page 5, line 19, the sentence now reads:**
**"The results also suggest that within plant isoprene oxidation is not the source of these VOCs in our experiments."**

p. 10, lines 4-5: the cited reference does not contain any information on the spectral peaks to monitor!
**This point has been corrected.**
**At page 10, lines 4-5, the sentence now reads:**
**"Details on the spectral peaks used to monitor each compound in red oak are reported in the Table S2 in Supplementary Information".**

We hope that the revised manuscript will be acceptable for publication in ACP. We thank you for your attention and look forward to hearing from you.

Sincerely,

Luca Cappellin

[revised manuscript text omitted]